# Fighting Phytopathogens with Engineered Inorganic-Based Nanoparticles

**DOI:** 10.3390/ma16062388

**Published:** 2023-03-16

**Authors:** Eirini Kanakari, Catherine Dendrinou-Samara

**Affiliations:** Inorganic Chemistry Lab, Chemistry Department, Aristotle University of Thessaloniki, 54124 Thessaloniki, Greece; kanakari4eirini@gmail.com

**Keywords:** nanotechnology, inorganic-based, metal-based, nano-agrochemicals, phytopathogens, anti-fungal, anti-bacterial, anti-viral, insecticidal

## Abstract

The development of effective and ecofriendly agrochemicals, including bactericides, fungicides, insecticides, and nematicides, to control pests and prevent plant diseases remains a key challenge. Nanotechnology has provided opportunities for the use of nanomaterials as components in the development of anti-phytopathogenic agents. Indeed, inorganic-based nanoparticles (INPs) are among the promising ones. They may play an effective role in targeting and killing microbes via diverse mechanisms, such as deposition on the microbe surface, destabilization of cell walls and membranes by released metal ions, and the induction of a toxic mechanism mediated by the production of reactive oxygen species. Considering the lack of new agrochemicals with novel mechanisms of action, it is of particular interest to determine and precisely depict which types of INPs are able to induce antimicrobial activity with no phytotoxicity effects, and which microbe species are affected. Therefore, this review aims to provide an update on the latest advances in research focusing on the study of several types of engineered INPs, that are well characterized (size, shape, composition, and surface features) and show promising reactivity against assorted species (bacteria, fungus, virus). Since effective strategies for plant protection and plant disease management are urgently needed, INPs can be an excellent alternative to chemical agrochemical agents as indicated by the present studies.

## 1. Introduction

Subsequent to water, genetic yield potential, and adaptation, crop losses due to pathogens, animal pests, and weeds are major yield constraints, responsible for losses ranging between 20% and 40% of global agricultural productivity [1]. Precisely, it is estimated that losses from phytopathogens such as bacteria, fungi, and viruses are increased by the high intensity of cultivation; at the same time, the yield-limiting potentials of pests, nematodes, and weeds could be reduced by 30% to 55% [2,3,4,5,6]. Reducing crop losses is an absolute priority given the increasing human population. The ultimate purpose of crop protection is not the elimination of pests or phytopathogens but to minimize crop losses to an economically acceptable level [4].

Conventional agrochemicals, including bactericides, fungicides, insecticides, and nematicides that are used to control pests and prevent plant diseases, are classified into several groups based on their structure and chemical composition [7]. Organophosphates, chlorinated hydrocarbons, carbamates, and carbamide derivatives are commercial agrochemicals’ most common active ingredients. These traditional formulations have a variety of limitations such as high organic solvent content, dust drift, long life in soil, and being released into the air; as a result only 1% are active on crops [7]. Besides the severe environmental pollution from pesticide overuse, phytopathogens develop resistance. To cap it all, there are harmful consequences on human health and animals via skin absorption and inhalation or changes in the level of antioxidant and oxidant enzymes in the human body [7]. Moreover, many of these formulations have been phased out of the market, and new classes of agrochemicals are unlikely to be available soon as it is a time-consuming, laborious, and costly process for companies to develop new ones, without certain results. For instance, the Environmental Protection Agency proposed that pesticides with glyphosate as one of their bioactive ingredients are restricted or banned because they can migrate and accumulate in the upper trophic levels of the food chain [8]. This impedes the commercial research and development of alternative phytoprotective agents. Therefore, new methodologies are needed to alleviate the serious pesticide contamination of the ecosystem. In this vein, an effort to improve agriculture by using nanotechnology and nanomaterials is under way.

Nanoparticles have gained recognition because particle size below 100 nm imparts new behavior and properties based on the large surface area and quantum effects. The recent emergence of nanotechnology in drugs and pharmaceuticals has opened up new opportunities to apply the fundamentals of nanotechnology to the agriculture sector [9]. Regarding pest control, crop productivity can be enhanced by introducing specific active substances in minimum concentrations at nanostructures to target specific sites. In particular, using compatible methods makes it possible to perform interventions to conventional pesticides without modification of their physicochemical and mechanical behavior, ensuring long-lasting effects [10]. There are several approaches which can be used to improve the efficacy of existing pesticides or to enhance their environmental safety profiles, or both, such as: *nanoemulsions* that increase the apparent solubility of poorly soluble active ingredients while keeping the concentration of surfactants lower than that in microemulsions; *nanoencapsulation* where pesticides are entrapped, as active substances, in various organic materials to form different sizes in the nano range in order to achieve controlled release [9]; water-insoluble *nanogels* that are less prone to swelling or shrinking with changes in humidity. These approaches can significantly improve the loading and release profiles of active ingredients [11].

Meanwhile, a strong focus on inorganic nanoparticles (INPs) has developed, especially those with bioessential metals such as Cu, Ag, and Zn. They are chosen for their broad range of antimicrobial activity against phytopathogenic bacteria, fungi, insects, nematodes, and viruses at relatively low doses that are well tolerated in plants and humans [12]. They have superior chemical and thermal stability compared to their organic counterparts, offering long-term efficiency; this means that they can be more easily stored, transported, and used in challenging environments and constitute a powerful tool to increase agricultural production and alleviate food insecurity [11,13]. It is important that a dual effect as fertilizers and antiphytopathogens can be supported. Moreover, the components are relatively cheap and easily integrated into more sophisticated structures such as nanomaterials and nanocomposites [13]. Indeed, the statement that “there is a far deeper understanding of the formation of inorganic particles than of organic particles” remains true today [14]. The use of these metals for their antimicrobial activity is not new, but nowadays, the advances in nanotechnology allow for the production of well-defined nanoparticles and the precise control of the physicochemical properties at the nanoscale leading to higher bioactivity than the corresponding bulk materials [15]. The size, shape, structure, and surface chemistry of INPs, known as 4S, govern their efficiency [16,17]. In particular, size and shape contribute to the NP’s ability for successful attachment and entrance inside the microbe cell [18,19]. Based on the inherent properties of the metals, different structures are isolated that can play a role in releasing ions, which will affect the toxicity mechanism against the microbe organism inside the plant [20]. Surface reactivity and surface coating give the proper charge to the INP, which in turn can react and bind to the target [18]. However, the significance of the anti-phytopathogenic activity of INPs depends on the different sensitivities of different microbe species, the duration of the INPs’ incubation, and the stage of treatment in the infected plants. The uptake pathway, foliar or soil sprayed, that the INPs will follow in plants seemed to affect their impact against microbes, according to their effective dose [18,19]. The proposed parameters that are involved in INPs’ effectiveness against phytopathogens are illustrated in Figure 1. 

Another challenging issue yet to be resolved includes the simple, successful, and possibly low-cost protocols for large-scale preparation and commercialization. Hence, there is limited progress in the evaluation of INPs in the agriculture sector as a way to reform modern agricultural practices. Ultimately, applications of these nanomaterials can add tremendous value in the current scenario of global food scarcity. Herein, given the multifactoriality, we focus on reviewing engineered/synthesized INPs that are well characterized (size, shape, composition, and surface features) and their reactivity against classified species (bacteria, fungus, virus) to assist future studies on (i) the development of new INPs, (ii) more complicated structures that are based on functional effective INPs, (iii) large-scale preparation of morphologically pure INPs and (iv) field experiment procedures. Biosynthesized INPs are out of the scope of the present review. We first describe mechanistic aspects and then review INPs that may be categorized as favored with intensive studies as well as those receiving less attention, based on the reported publications. The antiphytopathogenic behavior of advanced inorganic-based nanostructures are also considered. 

## 2. Mechanistic Aspects 

INPs have been used in several studies to determine their effects on a broad spectrum of phytopathogens and pests [21]. The physicochemical properties of INPs, such as their size, shape, surface charge, and chemical composition, can affect their diffusion inside plants [22,23]. The contact area between their surface and the cell membrane for better adhesion to the pathogen and even the stress-energy required to move the cell membrane upon entry of the nanoparticles are all relevant [22,23]. Before examining the interaction of INPs with plant pathogen species, it is crucial to refer to the mechanisms of uptake of nanoparticles by plants, which are related to the nature of the nanoparticles themselves, the physiology of plants, and the interaction of nanoparticles with the environment [24,25,26].

Generally, NPs can enter plant tissues through root tissues or above-ground organs/tissues such as epidermis, trichomes, stomata, stigma, and hydathodes, including wounds and junctions in roots (Figure 2B) [24,25]. The complexity of nanoparticles leads to different uptake mechanisms in leaves, roots, and other parts of plants, with a strong dependence on the characteristics of both the plant and the nanoparticles (Figure 2A) [24,25]. Thus, the nanoparticles move and internalize differently [27,28]. Movement in plant tissues is either by the apoplastic or the symplastic pathway and is a size issue (Figure 2C). During apoplastic movement, NPs move through extracellular spaces, cell walls of the adjacent cell, and xylem vessels [24,25]. The apoplastic movement allows the NPs to move towards the central cylinder of the root and the vascular tissues for further movement to the superficial part through the xylem following the transpiration system, and this route is preferred for larger NPs around 200 nm in size [25,26]. During symplastic movement, the NPs move between the cytoplasm of neighboring cells via plasmodesmata, which are tiny channels that cross the plant’s cell walls, and it is favored for NPs smaller than 50 nm (Figure 2D) [25,26]. Thus, large NPs tend to accumulate in the apoplastic space, whereas roots can take up small NPs through pores, approximately 5–20 nm in size, within the walls of epidermal root cells [25,26]. The size of 40–50 nm is the threshold for NPs to move and accumulate in plants. Both application routes, foliar and root spray, are common and used to introduce nanoparticles into plants (Figure 2B) [25,26]. After application to the foliage, the NPs follow a lipophilic or hydrophilic pathway to enter the plant system [25,26]. Lipophilic diffusion of the NPs takes place through the cuticle waxes of the leaf, while in the hydrophilic process, the nanoparticles move through the polar water pores present in the epidermis and stomata (Figure 2D) [25,26]. As the pores of the epidermis in the leaves are about 2 nm across, foliar application is size dependent [25,26]. 

Moving to the antimicrobial action, there are varied mechanisms by which INPs cause bacterial and fungus cell death or the inhibition of virus and insect feeding in plant organisms. Variations in susceptibility occur between species, different cell types, and growth conditions within the same species. The proposed modes of INPs’ activity against bacteria and fungi follow similar pathways, and their mechanisms of action against plant viruses and insects are given in brief. 

**Antibacterial** and **antifungal** activity of INPs is relatively more studied. Three modes of action lead to cell membrane rupture and subsequent bacterial or fungal cell death [29,30,31,32]. At first, the nano-size effect of INPs enables their deposition on the bacterial/fungal surface; the released metal ions interact with the bacteria or fungi, and ROS production induces oxidative stress and damage (Figure 3). 

The general architecture of bacterial and fungal cell walls appears to be conserved [33]. The bacterial cell wall consists of a peptidoglycan layer with an outer membrane of lipopolysaccharide molecules which carry a negative charge [34]. In a fungus, the inner cell wall is a chitin-glucan matrix, and the outer layer of the wall is rich in mannosylated glycoproteins [33]. These mannoproteins are linked to beta-glucans via glycophosphate groups that contain five mannose residues, and the phosphorylated mannosyl side chains give the fungal cell wall its negative charge [33]. Hence, the existence of negative anionic domains in the cell wall may increase the potential of metals to bind these structures and cause toxicity due to the relatively high INPs’ concentration [29]. 

In the presence of a large concentration of INPs, a focal source of continuously released ions penetrates the plant cells [29,30,31,32]. The adsorption of INPs/metal ions leads to cell wall depolarization, which changes the negative charge of the cell wall to become more permeable [29,30]. Thus, it is hypothesized that the positive charge of INPs and/or the released metal ions influence the interaction of the negatively charged cell wall of bacteria or fungi. The proposed high affinity increases the uptake of metal ions released due to INPs’ constant dissolution, causing intracellular damage [29]. The kinetic dissolution occurs faster when INPs are of smaller size (larger surface area to volume ratio) and have a rougher surface [29]. The antibacterial/antifungal activity of INPs can be proportional to the release of ions, which seems to be element-dependent phenomenon according to the higher oxidation susceptibility [29]. 

As INPs’ size is comparable to biological molecules, they may conceivably participate in subcellular reactions. Significantly, small INPs (large surface-to-volume ratio) are inclined to be more toxic, increasing ROS production and consequently inactivating a plant cell’s DNA, proteins, and lipid molecules [29,35]. The antibacterial and antifungal activity of INPs has been also linked to the stability of their planes [29,36,37]. Thus, the shape factor is essential as the planes with high atom density facets increase reactivity [29]. The increased abrasiveness (edges or defects) correlates with INPs’ toxicity because the increased surface area helps in adsorption and binding and directly affects the ROS generation [29]. ROS molecules are produced during the photocatalytic reaction, where oxygen enters undesired reduction states and transforms into free radicals, which are able to penetrate the plant cell membrane by lipid oxidation and cause cell death [29,31,32]. 

**Antiviral activity** of INPs in plants is still in its infant stage. Most plant viruses are rod-shaped with a capsid structure where protein discs form a tube (coat protein) surrounding the viral genome and rarely have an envelope [38]. It has been demonstrated that the charge is unevenly distributed on the surface of viruses belonging to different taxonomic groups. The negative charge is predominantly located at one end of the virus and is controlled by its genome [39]. The proposed pathways of action that INPs follow against plant viruses include two scenarios: the inhibition of virus from entering into plant cells or the inhibition of virus replication after entering into the plant cell [31,32,40]. ROS production is hypothesized to be the leading property which governs the above mechanisms, where reactive oxygen species may be increased due to the presence of released metal ions [38,41]. In the first place, INPs interact with the surface of the plant virus, thus inhibiting its entrance into plant cells [38]. Inside, the plant cell INPs bind to the virus’s coat protein, and due to INPs’ charge and high affinity to the virus genome, the replication is blocked [38]. It has been also stated that INPs activate the plant defense mechanisms such as the antioxidant system, resistance genes, and plant hormones that regulate plant protection against viruses [38]. The most crucial factor of INPs that influence the antiviral activity appears to be the treatment time, where better results are probably obtained when INPs are applied along with or after the virus infection [38].

**Insecticidal activity** of INPs in plants is also limited. However, some possible modes of action have been investigated. INPs, during their exposure to an insect, may follow the entrance via the mouthpart and digestive system (stomach poisoning), via fluid from a consumed host organism (inhalation poisoning), or via the epidermis upon contact [42,43]. The nano-size effect of INPs improves their dispersal and permeability, and thus increases the rate of entering in the insect body [42]. A widely accepted theory for INPs is that they achieve toxicity by triggering oxidative stress in insect tissues due to their penetration through the exoskeleton [43]. The INPs are able to bind sulfur from proteins or phosphorus from DNA in the intracellular space, leading to rapid denaturation of organelles and enzymes [43]. Oxidative stress is believed to be caused by the released metal ions. Specifically, in stomach poisoning, nano-induced oxidative stress occurs in the insect’s gut leading to a decreased activity of gut microflora due to epithelial cell damage and decreased extracellular enzyme activity [43]. In contact poisoning, INPs bind to the insect’s epicuticle (upper cuticle) owing to triboelectric forces, destroying its wax layer and lipids and resulting in the insect’s dehydration [43]. INPs can also cause trypsin inhibition, thus triggering insects’ inability to digest proteins and disrupting development [43].

**Figure 3 materials-16-02388-f003:**
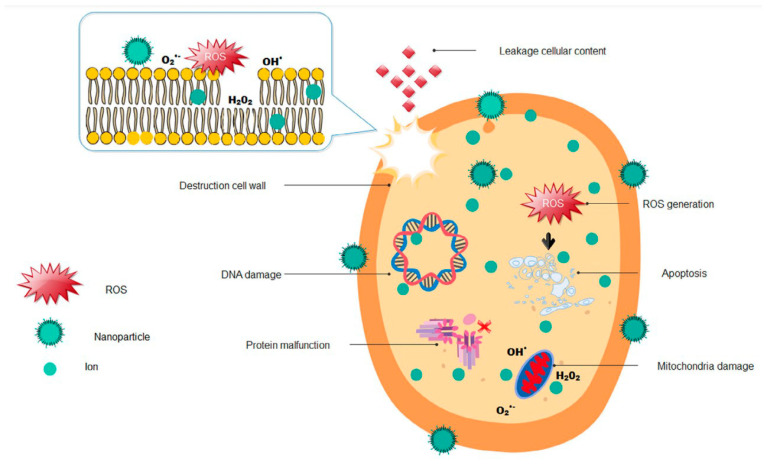
Schematic representation of antimicrobial actions of inorganic-based nanoparticles. Reproduced with permission from Ref. [35]. Copyright 2020 Wiley-VCH.

## 3. Favored INPs

Numerous metal nanoparticles have been synthesized and used to control phytopathogens. However, some of them have been intensively studied and shown outstanding properties to control pests and prevent plant diseases and thus are presented in priority (Figure 4). Considering the inherent properties of these elements, a general intro is given before each specific action as we believe this is helpful for a new researcher in the subject. Meanwhile there are several other metals and/or complicated structures that are promising but less investigated and are subsequently analyzed. 

### 3.1. Silver

Silver has for centuries accompanied humanity; for instance, Ag foils have been historically used to prevent infection of surgical wounds. In recent decades, the development and use of silver nanoparticles was inevitable. The antioxidant, antimicrobial, and anticancer properties in addition to the easy production, relatively low cost, and biocompatibility make nanosilver significantly attractive. As of 2025, it is expected that Ag NPs production will reach approximately 800 tonnes. Indeed, as compared to other nanoparticles, Ag NPs are thought to have a higher market value and commercial use, and they are widely advertised in consumer products such as Nano Green pesticide by Nano Green Sciences, Inc., as an innovative approach for controlling pests and phytopathogens [44,45,46]. 

Silver NPs are prominent against fungal species for plant disease treatment while they have also been applied to bacterial and virus phytopathogens. In general, the size, shape and surface reactivity of Ag NPs influence their mode of activity, but also, Ag^+^ ions, released from Ag NPs’ surface upon contact with water, play a major role [47,48]. The amount of Ag^+^ ions depends on the Ag NPs size and are following the trend to smaller size, since the smaller nanoparticles release many more Ag^+^ ions. Ag^+^ ions occur through the dissolution of one or two surface silver oxide monolayers [48]. Irregular pits on the cell wall of the phytopathogen can be formed by Ag NPs, promoting ions’ entrance to the cell, and according to a hypothesis, Ag^+^ ions may also enter the cell through cation-selective porins providing another possible mechanism of toxicity [29,49]. Further, Ag NPs and Ag^+^ ions react and bind to thiol groups in essential pathways such as respiratory and cell wall synthesis enzymes [49,50]. The protein-NP interaction in the SH^−^ group of the mannose phosphate isomerase causes an interruption of cell wall synthesis, leading to simultaneous leaching of internal components and cell death [29,49].The physical attachment of the Ag NPs to the DNA is also significant, probably due to the high affinity of Ag^+^ ions to phosphate groups in the molecules of the bacterial, fungal, or viral genome, causing denaturation of the DNA and interruption to cell division [49,50].

#### 3.1.1. Antifungal Effect

Six different *Rhizoctonia solani* anastomosis groups infecting cotton plants were studied in vitro and treated with Ag NPs (spherical, 40–60 nm), where the higher suppression of fungal radial growth was noticed with the increase of Ag NPs’ concentration at 0.0019 mol/L [51]. It is suggested that released Ag^+^ ions and Ag NPs mainly affected the function of membrane-bound enzymes, thus destroying the membrane integrity [51]. Likewise, Ag NPs that were stabilized with ammonia with an average size of 52 nm were tested against *Phomopsis* spp. in soybean seeds [52]. However, it has been stated that ammonia did not interfere with the antifungal activity [52]. By increasing their concentration (270–540 μg/mL), inhibition was increased due to the high density at which the solution saturates, coheres to fungal hyphae, and destroys the membrane integrity [52]. Much smaller Ag NPs (spherical, 5–24 nm) stabilized with gelatin as capping agents showed a fungistatic effect upon the phytopathogen *Colletotrichum gloesporioides*, that causes anthracnose in a wide range of fruits, in a dose-dependent manner [53]. The small size effect and the low concentration (56 μg Ag NPs/mL, in the growth medium potato dextrose agar-PDA, was the highest used) inhibited the fungus almost by 90% due to attachment and penetration through the cell membrane, causing spores’ death [53]. Despite the described influence of positively charged NPs against microbial cells, an example of negatively charged polyvinylpyrrolidon -PVP coated Ag NPs was found, in a dose-response manner (100 mg/L) [54]. The influence of the surface coating of Ag NPs such as mono citrate (MC-AgNPs), cetyl trimethyl ammonium bromide (CTAB-AgNPs), and polyvinylpyrrolidon (PVP-AgNPs) was seen as the basis for the antifungal activity against *Sclerotinia sclerotiorum*, suggesting that growth inhibition was higher in case of PVP-AgNPs [54]. Significant inhibitory effect was observed on sclerotia formation (number and weight of sclerotia); this is important as sclerotia remain in dead plant tissue and soil, and can survive for decades, threatening crops’ health [54]. Furthermore, during in vitro experiments, parameters such as incubation time and type of growth medium appeared to play a significant role in the antifungal activity of Ag NPs (around 20 nm) against *Fusarium culmorum* [55]. Specifically, addition of 2.5 ppm of Ag NPs, 24 h after incubating *Fusarium culmorum* spores in a potato dextrose agar-PDA medium, greatly reduced the number of germination fragments and sprout length relative to the control, thus inhibiting the germinating process [55].The spore viability, colony development, and mycotoxin production of *Fusarium* spp. were carried out with 30 nm Ag NPs and doses of 2–45 μg/mL at different exposure times (2–30 h) [56]. Overall, regardless of the fungal species, the inhibitory effect of Ag NPs increased with increasing doses, and when the contact time between Ag NPs and spores increased, the effective doses decreased [56]. At high exposure times (20−30 h), the three effective doses for the studied parameters ranged from 1–30 μg/mL for all the *Fusarium* spp. [56]. 

Y. Jo et al. tested the activity of Ag NPs and Ag^+^ ions on two plant pathogenic fungi, *Bipolaris sorokiniana*, and *Magnaporthe grisea*, where the application time at three hours before spore inoculation governed the inhibitory effect [57]. Comparing the two fungi species, the colony formation of *Bipolaris sorokiniana* showed a more significant reduction of 50% in Lolium perenne plants, due to proposed direct attachment and penetration in the cell membrane of spores [57]. The same group of researchers also investigated Ag NPs (7.5 nm) in managing *Gibberella fujikuroi* in rice seedlings [58]. Dose-response at 150 μg/mL of Ag NPs and exposure time of ≥10 min after the infection indicate that Ag NPs needed to attach to the surface of microbial cells and disturb their function, thus decreasing conidia viability by 50% and the colony-forming units on the seed surface [58]. 

#### 3.1.2. Antibacterial Effect

The greater interest in the potential application of Ag NPs for managing plant diseases of fungal origin rather than bacterial is explained by the necessity of more complex nanocomposites to inactivate phytopathogenic bacteria [59,60]. Thus, several studies introduce suitable stabilizers/surfactants for Ag NPs to achieve the desired antibacterial effect. Three different surfactants (pectins, sodium dodecyl sulphate, fructose) have been used to synthesize spherical Ag NPs (10–30 nm) and were evaluated against several genera of bacterial strains [59,60]. Ag NPs stabilized with sodium dodecyl sulphate (28.3 ± 11.7 nm) showed the highest effectiveness with their minimum inhibitory concentration effect at 0.75–3 mg/L due to the combined action of released Ag^+^ ions and Ag NPs themselves, which interact with functional groups on the bacterial membrane leading to cell wall deformation and collapse [59]. The antibacterial activity of Ag NPs stabilized with Tween 80 surfactant was evaluated against *Ralstonia solanacearum* and is associated with the dilution of Ag^+^ ions [61]. Tween 80 played an essential role in the positive surface charge and stability of the Ag NPs dispersion by forming low zeta potential and preventing particle aggregation; the NPs blocked DNA replication and inactivated protein function in bacterium *Ralstonia solanacearum* [61]. TEM analysis showed that the cell membrane became rough, the cell structure was looser, and the cytoplasmic density indicated cellular content leakage [61]. The in planta experiment (tobacco bacterial wilt) revealed 96.71% antibacterial efficiency after seven days of treatment [61]. By using a surfactant such as bovine submaxillary mucin in the chemical synthesis of Ag NPs their stability and adhesive characteristics were improved in combating bacterial strains in plants such as *Acidovorax citrulli*, which causes severe infection in melon seeds [62]. The minimal size (5–20 nm) and the low concentrations (6.7–13.4 mg/L) of Ag NPs played a role in their penetration inside the bacterial cells, disorganizing their shape and causing surface vesicles, while no colonies of the bacterial strain were formed in the treated germinating seeds [62]. 

#### 3.1.3. Antiviral Effect

Although relatively little is known about antiviral effects, the NPs’ size and the application time are perhaps the most influential properties that determine the entry into host cells and subsequently the interactions with biomolecules. The prophylactic and 24 h post-application were shown to give a better control of the virus disease in plants [40]. To counter potato virus Y (PVY), spherical Ag NPs (12 nm, at 0.1 μg/μL) were applied 24 h after inoculation, implying that treatment timing was significant for Ag NPs to fulfill their antiviral activity through Ag^+^ ions interactions with sulfhydryl groups in the viral nucleic acids [63]. In contrast, the application of Ag NPs along with salicylic acid (at 0.1 μg/μL) three or seven days before inoculation with PVY or tomato mosaic virus (ToMV) was found to decrease virus concentration and percentage of infection, meaning that resistance was already induced by salicylic acid and/or the entry of the virus into the vascular system of plants was inhibited [63]. When solely Ag NPs were foliar sprayed at 50 ppm, seven days before inoculation with PVY virus in tomato plants, there was still a reduction in the infection [64]. TEM analysis confirmed that Ag NPs bind to the virus’s coat protein, inhibiting its replication in host plants, and chemical analysis showed a systemic acquired resistance induced due to increased total soluble protein, peroxidase, and polyphenol oxidase activity [64]. Again, the application time of Ag NPs 24 h after PVY virus inoculation seems to succeed, given the higher inhibitory effect at 200 ppm dose against tomato spotted wilt virus(TSWV) and bean yellow mosaic virus (BYMV) respectively, owing to the early phase of viral replication [65,66]. The small size of these Ag NPs (12.6 ± 5 nm and 8.54 nm respectively) elucidate the dramatic entrance into the host cell and the viral genome by binding to viral nucleic acid, functionally essential proteins, and cellular factors during virus replication and affecting virus-vectors interactions [65,66]. The beneficial result of blocking the virus acquisition and transmission by aphids to healthy plants depends on Ag NPs, which interact with virus particles and provoke chemical and physical changes or even affect the feeding behavior of the insect [66].

### 3.2. Cu-Based NPs

The antimicrobial properties of copper in the bulk form have been exploited for thousands of years. Cu and Ag have been used for water sanitization and food preservation since the time of the Persian kings [67,68,69]. A mixture of copper sulfate and calcium has been used as a fungicide since 1882 in Bordeaux, France, and reduced the percentage of *Plasmopara viticola* fungi in grape plants [70]. Meanwhile, several copper compounds have been extensively applied in agriculture as fungistatic on grapes and potatoes. However, due to the low water solubility, large amounts have to be applied for effectiveness. So, copper-based nanoparticles (Cu-based NPs), such as metallic copper (Cu), cupric oxide (CuO), cuprous oxide (Cu_2_O) NPs, as well as composite structures of Cu/Cu_2_O, Cu_2_O/CuO, or core-shell NPs, have attracted attention in addressing different species of phytopathogens [20,70,71]. Very recently, several companies have offered products with Cu-based NPs to control phytopathogens in crops. These products are marketed as more environmentally friendly and effective alternatives to their traditional counterparts. An example is the NANO-Cu™ marketed by Bio Nano Technology as a fungicide and bactericide trade product [72]. 

Copper effectiveness is attributed to the ability of copper ions to easily interconvert between Cu(I)/Cu(II) by Fenton-like (2) and Haber–Weiss (3) reactions [73] and generate ROS molecules, leading to lipid peroxidation, protein oxidation, and DNA damage [71]. Cu NPs, when exposed to aqueous environments, are susceptible to oxidation and through dissolution Cu^+^ ions release from metallic Cu NPs [74]. These released cuprous ions can produce ROS through the well-known Fenton and Haber–Weiss reaction while they self oxidize to Cu^2+^ [74,75]. Cu is an essential element for maintaining homeostasis in organisms; however, Cu ions may cause toxicity once they exceed the physiological tolerance range in vivo [76].
Cu^2+^ + •O_2_^−^ → Cu^+^ + O_2_ (First step of catalytic cycle),(1)
Cu^+^ + H_2_O_2_ → Cu^2+^ + OH^−^ + •OH (Fenton reaction),(2)
•O_2_^−^ + H_2_O_2_ → •OH + HO^−^ + O^2^ (Net reaction, Haber–Weiss),(3)

Further, cupric oxide NPs (CuO) cause direct toxicity by activating ROS production, and the free radicals, such as O^2−^, OH, and H_2_O_2_, subsequently oxidize biological molecules that lead to substantial oxidative stress and cell death [77,78]. Cuprous oxide (Cu_2_O), a visible light active p-type semiconductor, exhibits outstanding photocatalytic activity in visible light through the generation of electron-mediated hydroxyl (OH˙) radicals [79]. Thus, beside the great diversity of sizes of Cu-based NPs, composition effect is also very critical and is still a matter of discussion. Nevertheless, pure monometallic Cu NPs are more bioreactive than oxides; thus, they are also predicted to be toxic to the plants besides their antifungal effect [70].

#### 3.2.1. Antifungal Effect

The metallic form has been tested in several studies either of naked or coated Cu NPs and under this notion are given below. 

Starting with naked Cu NPs (25 nm), these have been tested in vitro against seven fungal species (*Botrytis cinerea*, *Alternaria alternata*, *Monilinia fructicola*, *Colletotrichum gloeosporioides*, *Fusarium solani*, *Fusarium oxysporum* and *Verticillium dahliae*) and their fungitoxic effect was 10 to 100-fold more severe to spores than hyphae, illustrating the possible target of Cu NPs, during their mode of action [80]. The chitin content in a fungus spore wall is lower than in a hyphal wall, rendering the former more susceptible to heavy metals [80]. Use of Cu NPs (50 nm, spherical) against *Alternaria solani* in an infested tomato crop reveals that high doses (10 and 50 mg/L) not only did not induce toxic effects but also promoted the activity of antioxidant enzymes and non-enzymatic compounds in the leaves and fruits [81]: superoxide dismutase, ascorbate peroxidase, glutathione peroxidase, chlorophyll a and b, vitamin C, glutathione, phenols and flavonoids were all increased with the elimination of ROS molecules [81]. Similar sized Cu NPs (53 nm, spherical) achieved complete inhibition against *Aspergillus niger*, *Fusarium oxysporum* and *Phytophthora capsici* [82,83]; it was observed that lower concentrations (7.5 ppm) suppressed the disease after one day, whereas larger concentrations (30 ppm) obtained the same results after three days. Thus, different susceptibility of the above fungal species in the tested concentrations showed that the efficient dose and incubation time influence the maximum outcome in fungal growth inhibition [82,83]. Much bigger sizes of Cu NPs (345.1 nm) with polygonal shapes showed better antifungal efficiency (46%) against *Fusarium oxysporum* compared to spherical-shaped Cu NPs (278.1 nm) at the same concentration [84]. The polygonal shape exhibits a larger surface area to volume ratio, contributing to the antifungal efficiency. The large size of Cu NPs, comparable to the targeted fungus entities, was considered as increasing the probability of an effect [84].

A variety of organic molecules have been used to stabilize metallic copper. Thus, Cu NPs (5–10 nm by DLS, spherical), stabilized with animal protein, non-ionic polymer and ionic polymer on their surface, were applied in different concentrations (300–600 ppm) with particular success, without any toxic effect on olive plants [85]. Actually, the mycelial growth of *Fusicladium oleagineum* and *Colletotrichum* spp. was the most vulnerable developmental stage despite the short-term stability of Cu NPs after their dilution in tap water in agriculture [85]. The ultra-small Cu NPs (3–10 nm by TEM, spherical) were also capped with cetyl trimethyl ammonium bromide (CTAB), to avoid rapid oxidation [86]. The in vitro assessment showed that their tremendous surface-to-volume ratio played a significant role in their antifungal activity against several fungal strains (*Phoma destructiva*, *Curvularia lunata*, *Alternaria alternata* and *Fusarium oxysporum*) compared to the commercially available fungicide bavistin [86]. The same coating (CTAB) has been used in case of Cu NPs (20–50 nm, spherical) against the growth of different *Fusarium* species (*Fusarium equiseti*, *Fusarium oxysporum*, *Fusarium culmorum*) [87,88]. The well-known toxicity of CTAB promotes the maximum activity of Cu NPs. The in vitro inhibition was found in a concentration dependent manner; the most efficient dose (450 ppm) achieved almost zero increase in the diameter of the fungal colony [87,88]. 

Beside the effectiveness of Cu NPs several studies refer to oxides. For example, in tomato plants infested with *Phytophthora infestans*, pegylated CuO, Cu_2_O, and Cu/Cu_2_O NPs (11–55 nm, spherical) have been evaluated in field conditions [70]. Similarly, oleylamine coated Cu_2_O@OAm NPs (30 nm, spherical) and Cu/Cu_2_O@OAm NPs (170 nm, nanorods) was tested in vitro against the yeast *Saccharomyces cerevisiae* [89]. Cu/Cu_2_O@PEG 8000 NPs (42 nm, spherical) were also synthesized and examined in vitro against *Fusarium oxysporum* [90]. Among these three different bioassays, and apart from the size effect, dose and amount of the organic coating, the results revealed that the antifungal activity is strongly related to the oxidation state of copper, indicating their complicated mechanism pathway [70,89,90]. Specifically, Cu_2_O NPs showed the most potent action against the fungal species, attributed to the amount of released Cu(I) ions [70,89,90]. Cu(I) produces more hydroxyl radicals, which are the most reactive free radicals and react with lipids, polypeptides, proteins, and nucleic acids [90]. The Cu(I) species can bind to proteins due to its affinity to thiol groups and can chelate proteins; thus, Cu(I) is more toxic than Cu(II), and consequently, the composition phase of Cu_2_O with Cu(I) species possesses higher antifungal activity than CuO [90]. Additionally, in comparison with the commercial copper-based pesticide products (Kocide 2000, Kocide Opti, Cuprofix disperss and Ridomil Gold Plus), the low concentration of pegylated Cu_2_O NPs (0.1–0.5 mg/mL) was adequate to give rise to binding and degradation phenomena on the fungal surface without any permanent damage to the plants [70,89,90].

Alternatively, CuS NPs were found to be promising agents against *Fusarium* spp. The size and shape parameters mostly governed their antifungal effect, where granular-shaped and spherical-shaped CuS NPs (100 nm) succeeded in reducing the diameter of the fungus’ growth zone [91]. 

#### 3.2.2. Antibacterial Effect

In vitro tested, CuO NPs (5.23 ± 0.8 nm by HR-TEM, spherical), with zeta-potential −12.23 ± 0.9, were stabilized with an animal protein [92]. The bactericidal effect against the tested phytopathogens (*Agrobacterium tumefaciens*, *Dickeya dadantii*, *Erwinia amylovora*, *Pectobacterium carotovorum*, *Pseudomonas corrugata*, *Pseudomonas savastanoi* pv. *savastanoi* and *Xanthomonas campestris* pv. *campestris*) exhibited a dose-dependent response (1500 ppm), which was greater than the Kocide 2000 35 WG and independent of the bacteria species [92]. Cu_2_O@PEG 8000 NPs (16 nm, spherical) have shown a critical specificity towards Gram-positive bacterial strains (*Xanthomonas campestris*, *Escherichia coli*, *Bacillus subtilis*, *Bacillus cereus*, *Staphylococcus aureus*), exhibiting the lowest IC_50_ values (2.13–5.59 μg/mL) in vitro [93]. This behavior is attributed to the predominant nanosized composition effect, resulting in ROS production, lipid peroxidation, and, most importantly, DNA degradation in a dose-dependent manner [93]. A comparison between different compositions of coated Cu-based NPs such as Cu@Tween 20, Cu_2_O@Tween 20 and CuO@PEG1000, against *Erwinia amylovora*, *Xanthomonas campestris and Pseudomonas syringae* was undertaken [94]. The in vitro and *in planta* assessments showed that neither the type nor the percentage of surfactants affected the antibacterial activity of these NPs, which indicates also the susceptibility of the bacterial strain [94]. However, Cu@Tween 20 NPs (46 nm, spherical) were the most potent against *Pseudomonas syringae* in bean plants (pot experiments in greenhouse conditions) attributed to the bioreactivity of the metallic core rather than the oxides counterparts [94]. The Cu@Tween 20 NPs (200 μg/mL dose) was five times more effective than the conventional pesticide Kocide 2000 35 WG (1000 μg/mL), without any negative impact on chlorophyll content [94]. The released Cu^2+^ ions led to penetration and rupture of the bacterial membrane [94]. Still, it is critical to consider the antimicrobial effect of the Tween 20 surfactant and experimental evaluation is needed before making any conclusions [95]. 

Majumdar et al. confirmed the size and concentration-dependent antibacterial activity of uncoated Cu NPs (18–33 nm, by TEM) against Xanthomonas *oryzae pv. Oryzae* (Gram-negative) in infected rice seedlings. The increased ROS production and dissolved copper ions were responsible [74,96] for the antibacterial activity that was comparable and even better than the commercial pesticide of the Bordeaux mixture. Interestingly, their fate in the plant’s metabolic cycle was investigated through various copper-dependent enzymes like superoxide dismutase and proteins like plastocyanin [96]. An increase in the activity of these enzymes and proteins was proportional to the higher concentrations of Cu NPs, resulting in the hermetic effect, which shows the positive response of plant metabolism to small doses of stress-producing components [96].

#### 3.2.3. Insecticidal Effect

Although Cu-based NPs have attracted attention for their exciting antifungal and antibacterial effect in phytopathogens, only one example of chemically synthesized nano pesticide was noteworthy. Specifically, nanostructured CuO NPs (20 nm) with a flower-like morphology showed an immediate entomotoxic effect against cotton leafworm (*Spodoptera littoralis*) with an LC_50_ value of 232.75 mg/L after three days due to their physical characteristics and interfacial surfaces upon insect mid-gut and cuticle layer of insect body wall [97]. Indeed, CuO NPs exhibited mesoporous network architectures with pore diameters of 3.38 nm in their surface area; these structures may contributed to the adsorption of biomolecules (proteins, fats, and carbohydrates) of the insect exoskeleton, leading to cuticle abrasion, cell membrane damage, leakage of intracellular contents, deterioration of the protective wax layer based on ROS production, and eventually, insect death by dehydration [97]. 

### 3.3. Zinc Oxide NPs

ZnO is listed as a “generally recognized as safe (GRAS)” material by the Food and Drug Administration and is used as a food additive [98]. Moreover, ZnO NPs are used in sunscreens, toothpaste, anti-dandruff shampoos, anti-fouling paints, and other modern personal products [99]. ZnO nanostructures exhibit high catalytic efficiency and strong adsorption ability as the oxygen atoms in the ZnO lattice are oxidized by photogenerated holes when exposed to UV light, allowing the ZnO NPs to release Zn^2+^ ions into the aqueous solution [98,100]. Regarding agriculture and plant growth, Zn is a crucial component for plant growth, in low doses, because it is a catalytic and structural protein cofactor in many enzymes and has structural functions in protein domains that interact with other molecules [101]. 

Zn is a micromineral nutrient that can also improve nutrient use in plant breeding [100]. Additionally, ZnO NPs show significant antimicrobial activity; inherent characteristics such as particle size, concentration, morphology, and surface activity affect their modes of action, including their excellent photocatalytic property, through light irradiation and zinc ions (Zn^2+^) release in the medium [99,100]. The electrostatic interactions between ZnO NPs and microbial cell walls destroy cell integrity and the liberation of Zn^2+^ ions [100]. At the same time, in the presence of illumination, oxygen molecules are desorbed from the active surface of ZnO NPs, and a series of ROS (H_2_O_2_, O_2_^−^, OH^●^) are formed on the surface of the ZnO nanocrystal [35,99,102]. The solubility of ZnO NPs may be another critical factor that influences their antimicrobial properties [100]. In general, the proposed antimicrobial mechanism of ZnO NPs, which govern the different effects against phytopathogens, consists of the disruption of cellular structure, the inhibition of protein and enzyme activity, the prevention of DNA replication, and the destruction of targeted antioxidant systems through ROS and Zn^2+^ [100]. Interestingly, in terms of raw materials, Zn is more abundant than Cu, and for that reason, the cost of Zn-based nanopesticides can be lower than that of Cu-based nanopesticides, considering the production of such products on a large scale [100]. 

#### 3.3.1. Antifungal Effect

Spherical ZnO NPs 20–35 nm (9–12 mmol/L) have been tested against the coffee fungus *Erythricium salmonicolor;* the generated ROS and Zn^2+^ ions impacted the function of N-acetylglucosamine or b-1 3-D-glucan synthase [103]. N-acetylglucosamine synthesizes chitin (a polysaccharide of great importance in the structure of the cell wall), while the b-1 3-D-glucan synthase participates in the synthesis of b-1,3-D-glucan (another essential component of the cell wall in fungi); thus, the fibers of the hyphae were noticeably thinner and tended to clump resulting in detachment of the cell wall [103]. Similar size and shape of ZnO NPs (30 nm, spherical) have been applied in wheat plants by foliar application at the anthesis stage; this was found to control the *Fusarium graminearum* and deoxynivalenol (DON) formation [104]. Analysis results from harvested wheat grains indicated that ZnO NPs reduced the number of fungus colonies and the toxin (DON formation) to non-detectable levels while the Zn residues remained at the internationally recommended levels for consumption [104]. Thus, the most critical application value of ZnO NPs lies in blocking and inhibiting the synthesis of secondary metabolites known as mycotoxins [99]. Further, Dimkpa et al. tested the synergistic effect of ZnO NPs with a biocontrol bacterium on reducing plant pathogen *Fusarium graminearum* [105]. This point of view revealed no synergism in mung bean broth agar between ZnO NPs and biocontrol Pseudomonas chlororaphis O6, but still, significant dose-dependent inhibition of fungal growth was observed due to the produced Zn^2+^ ions [105]. Furthermore, the shape-dependent antifungal activity of ZnO in the form of nanoparticles, lamellar platelets and hexagonal rods was studied in vitro against *Fusarium* spp. and *Colletotrichum gloesporioids* [106]. The platelet-shaped particles (average diameter around 246 ± 40 nm with an average thickness of 48 ± 6 nm) had better antifungal efficiency, and specifically, the growth of *Fusarium solani* was reduced by up to 65%, which means that the interaction between the particle and the fungi is selective [106]. The proposed mode of action was attributed to the different contact faces with the fungi and/or the internalization of particles into the cell by different routes depending on the nanoparticle structure and cell type being promoted by the corona protein formation [106]. In case of ZnO NPs (20–70 nm), the duration of treatment was found to have a synergistic effect that influences the combat against the fungi *Colletotrichum* sp. by causing loss in the continuity of some hyphae and the formation of groups of hyphal structures [107]. Specifically, by increasing the dose of ZnO NPs up to 15 mmolL^−1^ for 6 days there was a tremendous 96% inhibition of the fungal growth [107]. 

Considering the structural defects of ZnO NPs (<100 nm, spheroidal), it is also important to highlight that hydroxyl groups on the surface contribute to their antifungal activity where thinning of the fungal cell wall and lack of fibrillar network occur [108]. The area of halo of inhibition in cultures of *Mycena citricolor* revealed a 93% inhibition of growth (at 9 mmol·L^−1^) and the absence of reproductive structures (gems) [108]. The underlined mechanism of ZnO NPs refers to the oxidation of the proteinic corona that is formed on the fungal surface; thus, these oxidized proteins would serve as a cellular signal of oxidative stress for the cell wall [108].

An irregular shaped and aggregated porous structure of 65.3 nm average-sized ZnO NPs confirmed the pattern of shape-dependent potent behavior, where Zn^2+^ showed effective postharvest disease control against different filamentous fungi [109]. ZnO NPs/Zn^2+^, along with ROS molecules, interacted with the fungal cell wall and accumulated in the cytoplasm causing cell metabolism disturbances, impairment of the nucleic acid material by their irreversible adherence, ribosome disassembly, protein denaturation and electron chain disruption [109]. Interestingly, the doping of ZnO nanosized structures (55–100 nm) with Pd or Ce advanced their antifungal activity against several fungal species, because of the morphology (enlarged specific surface area in flower-like shape) and the increased concentration used in the assays [110,111]. These noble metals such as Pd and Ce are highly active and may change the surface properties of ZnO nano structures by giving them more negative charge which in turn results in their better dispersion and more production of reactive oxygen species [110,111].

#### 3.3.2. Antibacterial Effect

Relatively little is known considering ZnO NPs’ behavior against phytopathogenic bacteria. However an antibacterial screening (*Pseudomonas syringae*, *Xanthomonas campestris*, *Pectobacterium carotovorum*, *Pectobacterium betavasculorum*, *Ralstonia solanacearum*) with ZnO NPs (<100 nm) assessed their combined action of increased plant nutritional status and bacterial disease suppression in tomato and beetroot plants [112,113]. The maximum reduction in bacterial diseases was dose-dependent (200 mgL^−1^) and the foliar spray was the most effective method of treatment [112,113]. ZnO NPs reacted with H^+^ ions to produce H_2_O_2_ that can penetrate the bacterial cell membrane, leading to a continual release of membrane proteins and lipids, which changes the permeability of the cell membrane and thus causes cell lysis [112,113]. Thus, it is proposed that the enhanced plant growth (photosynthetic pigments, proline content) acted in a complementary manner with the reduction in bacterial disease indices. 

Nanorods of ZnO NPs were foliarly applied in tomato plants to control the disease bacterial speck caused by *Pseudomonas syringae* [114]. By the in vitro evaluation, a concentration-dependent antibacterial activity was found that was connected to the oxidation of glutathione by free radicals that destroyed the cell membrane and induced deformation of the contents of the cytoplasm, leading eventually to cell death [114]. The pot experiment in greenhouse field conditions also revealed a resistance induction mechanism where pathogenesis-related genes (LePR-1a, Lipoxygenase) and self-defense enzymes like peroxidase and polyphenoloxidase were highly detected in treated plants compared to the untreated ones [114]. 

#### 3.3.3. Antiviral Effect

Cai et al. illustrated an example of systemic resistance induction to investigate ZnO NPs’(55 nm, spherical) antiviral activity against the Tobacco Mosaic Virus (TMV) in Nicotiana benthamiana plants [115]. The daily foliar spray of ZnO NPs onto tobacco leaves for 12 days induced direct suppression, attributed to the injury of virus shell proteins, preventing viral entry and replication inside the host plant [115]. The mode of action that lies behind the antiviral activity consists of ROS accumulation, up-regulation of peroxidase, catalase activity, systemic resistance-related genes, and increased phytohormones levels like SA (162%) and ABA (517%) [115]. Similar triggering of the antioxidant defense system, in tomato plants, happened when ZnO NPs (100 mg/L), were foliarly sprayed to mitigate the adverse effects caused by Tomato Mosaic Virus (ToMV) infection [116].

## 4. “Less” Studied INPs

In addition to the above structures, there are “less” studied INPs with less frequent appearance in the literature. For instance, pegylated Ca(OH)_2_ NPs(16.5 ± 0.15 nm, TEM) were evaluated against second-stage juveniles of *Meloidogyne* spp. [117]. It was revealed that the release of [OH]-anions boosted Ca(OH)_2_ NPs’ nematicidal efficiency as the transport of anions happens through the ion canals of the root-knot nematodes [117]. Nevertheless, the following INPs present equally remarkable anti-phytopathogenic effects.



*Antifungal effect*



**Iron-based NPs**: The antifungal activity of Fe_2_O_3_ NPs (10–30 nm, spherical) against several fungi species (*Trichothecium roseum*, *Cladosporium herbarum*, *Penicillium chrysogenum*, *Alternaria alternata* and *Aspergillus niger*) was studied [118]. The results showed that by increasing the concentration of Fe_2_O_3_ NPs, there was the highest inhibition in spore germination against *Trichothecium roseum* (87.74%) and the highest zone of inhibition against *Penicillium chrysogenum* (28.67 mm) with an activity index of 0.81; the MIC value range was 0.063–0.016 mg/mL for the different fungal pathogens [118]. Besides the oxidative stress induction and metal ion release, it was proposed that Fe_2_O_3_ NPs reduced oxygen supply for respiration [118]. Spherical CoFe_2_O_4_ and NiFe_2_O_4_ NPs (25 nm) were tested in a pot experiment against *Fusarium oxysporum* in capsicum seedlings; by increasing the concentration up to 500 ppm, there was a complete reduction of the disease as the mycelia growth was suppressed [119]. 

**Al-based NPs**: Spherical-shaped Al-based NPs (100–250 nm) with worm-like mesopore structures were tested against *Fusarium oxysporum* causing root rot disease in tomatoes [120]. In this morphology connected to the large regions of the Al NPs domains, Al and Al oxide domains exist and the released aluminum ions react with the thiol groups (-SH) of the proteins in the fungus [120]. The highest fungal growth inhibition was found at 400 mg/L without any phytotoxicity against tomato plants [120]. The oxidation stress in the fungal cell wall occurs through electrostatic attraction between the mesoporous Al-based NPs (positive charge) and the fungal cell [120]. The most significant feature of mesoporous Al-based NPs is the existence of plenty of small pore sizes as active sites for contact with the cells, which render cytotoxic effects against root rot fungus; thus, fungus does not easily become resistant compared to chemical fungicides that have only one target site [120].

**Si-based NPs**: Aggregated mesoporous Si NPs (20–150 μm) were tested in vitro and under controlled conditions (greenhouse) against the early blight of tomatoes caused by *Alternaria solani* [121]. Antifungal efficiency (400 mg/L) was attributed to their morphology and surface reactivity (cylindrically-shaped and uniform pore sizes) as well as the conversion of monodispersed meso-/macro-porosities into ultra-or micrometer-sized particles [121]. The attention to mesoporous Si NPs is ascribed to surface silanol groups (Si-O-H) and their unique characteristics, such as uniformed mesoporous tunnels, narrow pore size distribution, good biocompatibility, low toxicity, and chemical stability [121]. The antifungal effect has been ascribed via the facile breakdown of the cell wall due to the formed hydrogen bonds between lipopolysaccharides of the cell wall and surface hydroxyl groups present in mesoporous Si-based NPs [121]. Moreover, negligible phytotoxicity was observed in tomato plants, while the growth parameters were already significantly increased compared to untreated controls [121]. 



*Insecticidal effect*



**Al-based NPs**: The potential of Al oxide NPs (10μm, amorphous) as insecticide agents was evaluated against leaf-cutting ants *Acromyrmex lobicornis* and the major pest enemies in stored food supplies, *Sitophilus oryzae* and *Rhyzopertha dominica* [122,123]. Dry dust applications in treated wheat caused mortality by increasing the time of exposure and the concentration, where LC50 values were 127 and 235 mg/kg respectively [122,123]. Interestingly, the Al oxide NPs revealed enhanced attachment to the cuticle of exposed insects due to more excellent sorptive properties, but further experiments need to be done to identify the mode of action and their non-target toxicity [122,123]. 

**Si-based NPs**: The management of stored-grain pests such as *Callosobruchus maculates* and *Sitophilus oryzae* has been evaluated with SiO_2_ NPs (spherical, 20–60 nm) [124,125]. Indeed, silica-based inert dust has been increasingly used as stored grain protectant. Specifically, the mortality of adults (>80%) increased with increasing SiO_2_ NPs’ concentrations, and the effective doses ranged from 1 to 2.5 g/kg in cowpea or rice grains [124,125]. SiO_2_NPs did not affect the looseness and bulk density of grain mass even with the highest applied dose in the bioassays, while the insect mortality was attributed to the impairment of the digestive tract or to surface enlargement of the integument as a consequence of dehydration or blockage of spiracle and trachea [124,125]. 

**TiO_2_ NPs**: TiO_2_ needle-shaped NPs (with a diameter of 76.15 nm long and 8.52 nm wide) showed an insecticidal effect on *Bactericera cockerelli* second-stage nymphs under greenhouse conditions in tomatoes [126]. The direct foliar spray application to the plants in the greenhouse resulted in 32% mortality, although the in vitro results showed enhanced mortality (99%) in concentrations above 100 ppm [126]. This evidence was explained by the size of the plants, which were already large (40–45 cm approximately) with a large amount of foliage, so the low volume application (25 mL per plant), was inadequate to cover the foliage sufficiently [126].



*Antibacterial effect*



**TiO_2_ NPs**: Disease management practices often include strict sanitation and bactericide application in commercial floral crop production. TiO_2_ NPs have shown exceptional antibacterial activity against *Xanthomonas* spp. on geranium and poinsettia under greenhouse conditions [127]. The foliar application of the most effective dose (75 mM) presented a 67–93 % lower number of leaf lesions than untreated plants. At the same time, no phytotoxicity was observed, but further investigation is needed to fine-tune the concentrations and application times [127]. 



*Antiviral effect*



**TiO_2_ NPs**: TiO_2_ NPs coated with oleic acid feature hollow shape stacked nano-sheets that were formed through hydrogen bonding [128]. These micron aggregated structures (TiO_2_ 3–5 μm) present perforations, like a birdcage, which increased their exposed interaction area with the plant virus (broad bean strain virus, BBSV) in faba bean plants [128]. The decrease in the severity of the disease resulted from an up-regulation of the expression of PR-gene through the involvement of the salicylic acid signaling pathway, thereby perhaps blocking the interaction of the virus with the cell [128]. The foliar spray was found to be more effective due to the direct and fast contact with the plant virus [128]. 

## 5. Advanced Inorganic-Based Nanostructures

With the progress of nanotechnology, more complicated nanostructures are synthesized with more specific properties as compared to their single/individual counterparts (Figure 5). Generally, these structures can be categorized as inorganic–inorganic, organic–inorganic, and bio–inorganic hybrids with advanced properties. These second-generation structures have shown improved benefits mainly due to their distinctive material properties. However, the derived nanostructures in many cases exhibit improved properties not observed for any of the individual components. The development of novel nanocomposites for improved and/or specific management of phytopathogens can mitigate the emergence of resilient and persistent pathogens. Herein, we focus on the advanced inorganic-based nanostructures and their efficacy against phytopathogens. As yet, there are few examples in the literature in the agrochemical sector and further research is needed to fully understand their potential as a tool to control phytopathogens.

**Bimetallic nanostructures**: Bimetallic compositions such as alloys are more stable than their individual metal components, making them more durable and long-lasting as antiphytopathogenic agents. Their biocidal activity is enhanced due to the synergistic effect of the two different metals. Nanobrass, CuZn NPs and glycol-coated CuZn nanoflowers have shown vigorous fungicidal activity against *Saccharomyces cerevisiae*, *Botrytis cinerea*, and *Sclerotinia sclerotiorum,* respectively [129,130]. Specifically, CuZn NPs (20 nm) and CuZn-DEG nanoflowers (consisting of NPs with an average size of 35 ± 1.2 nm) demonstrated, in both cases, a dose-response antifungal efficiency by promoting reactive oxygen species [129,130]. The results indicated no phytotoxic effects observed during the pot experiments while the photosynthetic parameters were enhanced [129,130]. Pegyllated CuFe NPs (40 nm) were tested against the root-knot *Meloidogyne* spp. and exhibited the lowest EC50 value at 0.03 μg/g soil [131]. Nematicidal activity was attributed rather to the release of Cu ions than Fe ions. Additionally, fertilizing properties were indicated as the fresh shoot and root weight were increased in treated plants [131].

**Core-shell structures**: A core-shell structure consists of an inner core-metal and a monolayer or multilayers of a different shell-metal. These particles have been of interest as they can exhibit unique properties coming from the combination of the different core/shell metals, the thickness of shell, the geometry and design while it is possible to control a slow release to mitigate the probable phytotoxicity of each nano-metal component [132,133]. A silica core-shell composition with core diameters from 50 to 600 nm and ultra-small Cu NPs (<10 nm) and quaternary ammonium (Quat) molecules on the shell was tested against the bacterial spot disease in tomatoes [133]. This complex structure was shown to ameliorate the wetting properties with contact angles below 60° and use an adequate ratio of Cu to Quat, 4:1 mg/mL, for sufficient inhibition of the growth of *Xanthomonas perforans* which is Cu-tolerant [133]. Specifically, this design provided a slow ionic release of Cu to the leaves upon water washes without any sign of phytotoxicity even at the dose of 1000 μg/mL [133]. Meanwhile, the presence of Quat promotes the membrane permeability in the bacterial microorganism [133]. The field experiments revealed that the silica core-shell with Quat agents and 100 μg/mL Cu achieved inhibition of disease progression comparable to 200 μg/mL of the commercial Kocide 3000 [133]. In another multimodal approach, Cu-loaded silica gel matrix with dispersed ZnO rods (600–1100 nm), revealed strong effectiveness in controlling citrus canker disease in grapefruit trees [134]. This composite core-shell structure (ZnO–nCuSi) was assumed to present synergistic antimicrobial effect by two actions: from zinc oxide through oxidative stress and copper toxicity that targets protein inactivation, DNA damage and loss of membrane integrity [134]. Interestingly, ZnO–nCuSi was effective for two consecutive years’ field efficacy at less than half the metallic rate of the commercial cuprous oxide/zinc oxide pesticide [134].

**Doped nanostructures:** The doping approach, when two different semiconductors are combined, is considered the most efficient and stable way of building photocatalysts with enhanced ionic release and ROS production [135,136,137,138]. The visible-light photocatalytic performance can be improved and the photo-corrosion is inhibited. The charge carrier transfer between the two semiconductors is accelerated, enabling the effective separation of photo-induced electron-hole pairs, based on the synergistic effect of the hybrid photocatalysts [135]. 

Different dopants have been used for TiO_2_ NPs. The antifungal activity of Ag-doped hollow TiO_2_ NPs under visible light exposure was confirmed against *Fusarium solani* in infected potatoes. Ag doping promotes Ag-S and disulfide bonds formation in fungus cellular proteins (respiratory enzymes), leading to cell damage [137]. This mode of action includes generation of •OH radicals, which results from the oxidation of the surface water molecules by holes and the recombination of the electron-hole pairs; thus, the doping of Ag to hollow TiO_2_ NPs reduces the recombination rate by accepting the photoinduced electrons and holes as electron-hole traps, and increases •OH radical generation [137]. Meanwhile, light intensity and exposure time controlled the production of toxic naphthoquinone pigment significantly [137].

Another example is the Zn-doped TiO_2_ NPs, where this light-activated nanostructure showed antibacterial activity against *Xanthomonas perforans* in a time-dependent manner and dose-dependency of illumination, in controlling the bacterial spots of tomatoes [138]. Notably, 20 min of photocatalysis achieved in vitro inhibition of bacterial growth; in greenhouse conditions, ≈500 to 800 ppm of Zn-doped TiO_2_ significantly reduced bacterial spot severity, without any adverse effect on tomato yield [138]. 

Beside single metal dopants, inorganic compounds have been used to dope TiO_2_ NPs. For example, Ag_3_PO_4_ and Cu_2_(OH)_2_CO_3_ linked with visible-light-driven TiO_2_ have been synthesized to manage *Fusarium* spp. diseases in crops [135,136]. Liu et al. studied these hybrid photocatalysts TiO_2_/Ag_3_PO_4_ (TiO_2_ microspheres doped with 2–5 nm sized Ag_3_PO_4_ NPs) and TiO_2_/Cu_2_(OH)_2_CO_3_ (20–50 nm sized TiO_2_ NPs doped with Cu_2_(OH)_2_CO_3_ clusters with size of 2–5 nm) as an alternative method to fight pathogenic fungi *Fusarium graminearum* [135,136]. A significant reduction in the survival ratio of fungus macroconidia and complete inactivation was achieved in 80–100 min, and was attributed to cell wall/membrane damage by ROS molecules (∙OH and O^2−^) [135,136]. The photocatalytic disinfection mechanism behind the above doped structures is supported by their substantial oxidation power and higher O_2_ production under visible light irradiation [135,136]. 

Cu-doped ZnO NPs revealed a growth inhibition of fungi *Botrytis cinerea* and *Sclerotinia sclerotiorum* and nematode paralysis of *Meloidogyne javanica* in a dose-dependent manner [139]. Cu-doped ZnO were more effective against *M. javanica* (EC_50_ = 2.60 μg/mL) than the pegylated Cu NPs; the antifungal activity was approximately similar for both NPs, with EC_50_ values at 310 and 327 μg/mL against *B. cinerea*, respectively, and 260 and 278 μg/mL against *S. sclerotiorum*, respectively [139]. The treatment of lettuce plants with Cu-doped ZnO NPs increased the leaf net photosynthetic value at 4.60 and 6.66 μmol CO_2_ ^−2^ s^−1^ in plants inoculated with *S. sclerotiorum* and *M. javanica*, respectively [139].

**Nanocapsule formation**: Encapsulation is an adequate method to overcome issues coming from the instability and volatility of the essential oils (EOs) and/or protecting active ingredients for improving their distribution and controlled release. Inorganic-based nanocapsules of *Zataria multiflora* essential oil and ZnO NPs (ZnO-ZmEO) were investigated against six isolates of *Fusarium* [140] and *A. solani* [141], respectively. The mycelial growth inhibitory effect, in both individual investigations, was increased (by 42.70% compared to ZnO NPs and by 66.33% to EO). 

**Hetero-nanostructures**: Inorganic-inorganic composites can be formed by combining different inorganic features of NPs such as magnetic and plasmonic. In that vein, spherical Cu_2_O NPs (30 nm) with improved antifungal properties were functionalized with spherical NiFe_2_O_4_ NPs (9 nm), and this hetero-nanostructure system induced a magnetomechanical cell-stress in yeast *Saccharomyces cerevisiae* [142]. The magnetomechanical cell stress induction was accompanied by alteration of cellular membrane integrity and programmed cell death signaling. Thus, the low-frequency magnetic field promoted the antiphytopathogenic behavior [142]. Table 1 summarizes all the studied engineered INPs, by giving the nanoparticle properties and antimicrobial properties, respectively.

## 6. Conclusions and Future Outlook

Plant disease occurrence is complicated; it is based on the triangular relationship in which plant disease results only in the presence of an infectious pathogen, a susceptible host, and a disease-friendly conductive environment and their interactions. The indiscriminate usage of synthetic pesticides has created several problems such as environmental pollution, ecological imbalances and diseases in humans and animals. Moreover, there is no doubt that phytopathogens have increasing resistance to synthetic pesticides. Nanoagrochemicals constitute an alternative solution. Inorganic-based nanoparticles are being extensively exploited in the agrochemical sector and research is proceeding to optimize their synthesis, improve their stability and efficacy against phytopathogens, and reduce potential toxicity to non-target organisms. In that vein, a detailed analysis of the ongoing progress on the application of INPs for controlling phytopathogens in agriculture was presented. Their market has greatly increased over the last few years and is not expected to decrease. However, the need for discovery of less toxic and environmentally acceptable substitutes for commercial agrochemicals is amplified, creating a significant market opportunity for alternative and novel products. 

According to the literature review on engineered INPs, regarding their main characteristics (Table 1) there is not yet a perfect size, shape, or composition, and it appears that multi factors influence their behavior against phytopathogens. However, size and shape are two factors that must be adjusted in balance to achieve effective toxicity. In general, the smaller the size of the nanoparticles and the rougher their surface, the greater the chemical affinity with the plant pathogen. The antimicrobial activity of any INP’s composition follows the corresponding ionic release. The kinetic dissolution of INPs is fast, especially when the NPs are bare, and consequently the released metal ions result in increased uptake. Τhe choice of the organic coating and therefore the surface charge of the nanoparticles plays a decisive role in the contact with the plant pathogen as well as in the interaction with and adhesion to the target surface. The more effective dose seemed to be lower when applied foliarly in contrast with soil drench. Beside the above, microbe species placed challenges in choosing the ideal duration of INPs’ incubation and stage of treatment in the infected plants, where the prophylactic application resulted in the up-regulation of plant defense mechanisms. Among the advantages of INPs is that they can target specific phytopathogens and have a long residual effect, reducing the need for multiple applications; many of them are also micronutrients and beneficial for plant growth. 

Among the most studied INPs, Ag and ZnO NPs have been mostly studied for their anti-fungal effect, while Cu-based NPs have been explored to the same extent as both antifungal and antibacterial agents in plant diseases. In Ag NPs, beside the size and dose, the exposure and the application time influenced their effectiveness. The oxidation state (composition effect) and the high photocatalytic activity of Cu-based NPs and ZnO NPs, respectively, was the key factor which governed their anti-phytopathogenic behavior. Amongst the “less” studied INPs, the photocatalytic effect of TiO_2_ and mesoporous Si-based and Al-based NPs stand out. Advanced inorganic-based nanostructures (with incorporation of individual and functional INPs) constitute a new strategy against phytopathogens. Every so often they give rise to synergetic effects and they seem to have promise for multitarget effects but more research is needed for their further development. A definite drawback is that most of the studies were evaluated in vitro and/or in pot experiments. Therefore, more experiments in field conditions are required to evaluate the potential ecological impact of INPs on the environment’s biodiversity.

As yet, several obstacles need to be resolved before their “real-life” applications in sustainable agriculture, such as stability and aggregation of NPs, size distribution, control of crystal growth and sparseness of field experiments. In the future, there will be a need for low-cost protocols so that large-scale production of such nanostructures be successful for commercialization. Importantly, the potentially toxic effects on the environment and consumers’ health should be addressed to propose a holistic and safe approach to crop production. 

## Figures and Tables

**Figure 1 materials-16-02388-f001:**
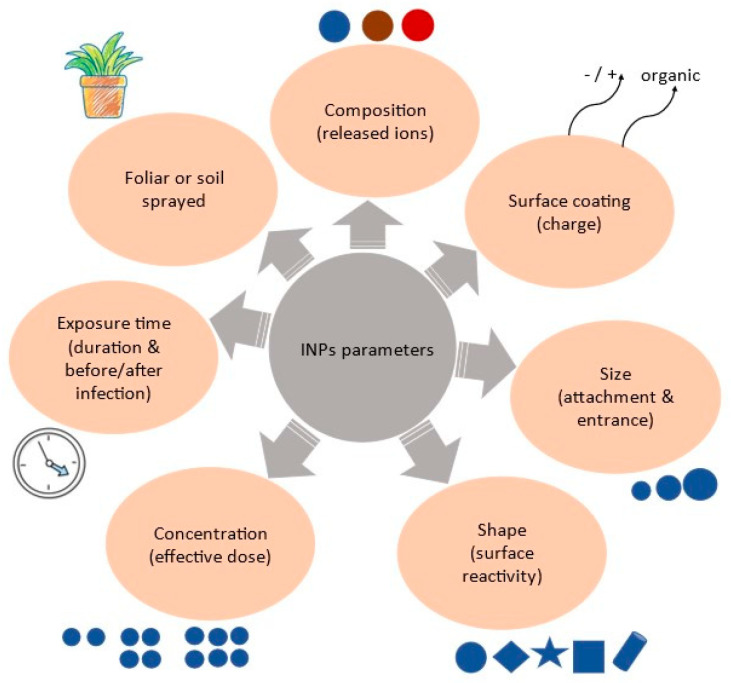
Schematic presentation of the INPs’ parameters that influence their performance as anti-phytopathogenic agents.

**Figure 2 materials-16-02388-f002:**
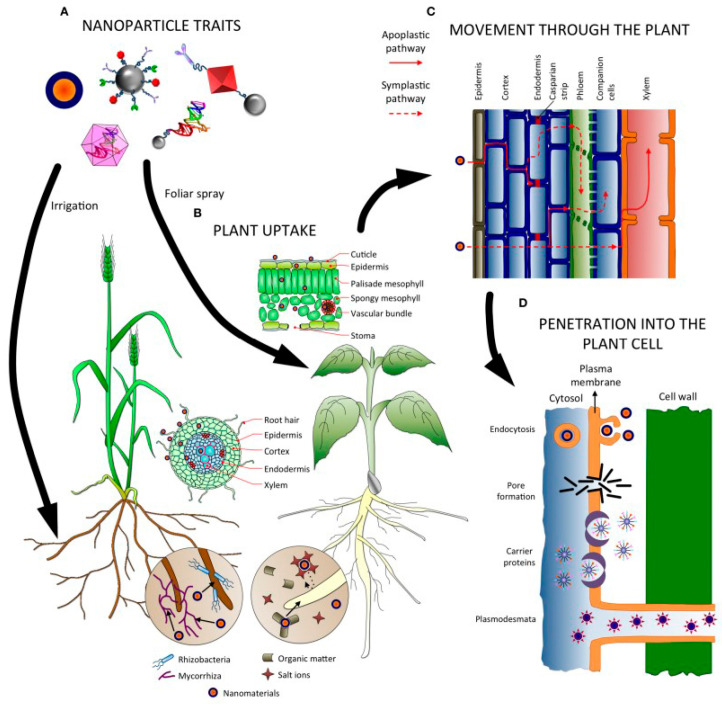
Schematic representation of the uptake mechanisms of NPs by plants. (**A**) Different composition effects of nanoparticles influence their uptake mechanism and their movement through plant tissues and cells. (**B**) Possible plant uptake pathways. (**C**) Two scenarios occur in which nanoparticles move through the plant (apoplastic and symplastic pathways). (**D**) Interaction of nanoparticles with plant cells for entering plasma membrane through pores. Reproduced from Ref. [27] under License CC BY Copyright (2017), Frontiers.

**Figure 4 materials-16-02388-f004:**
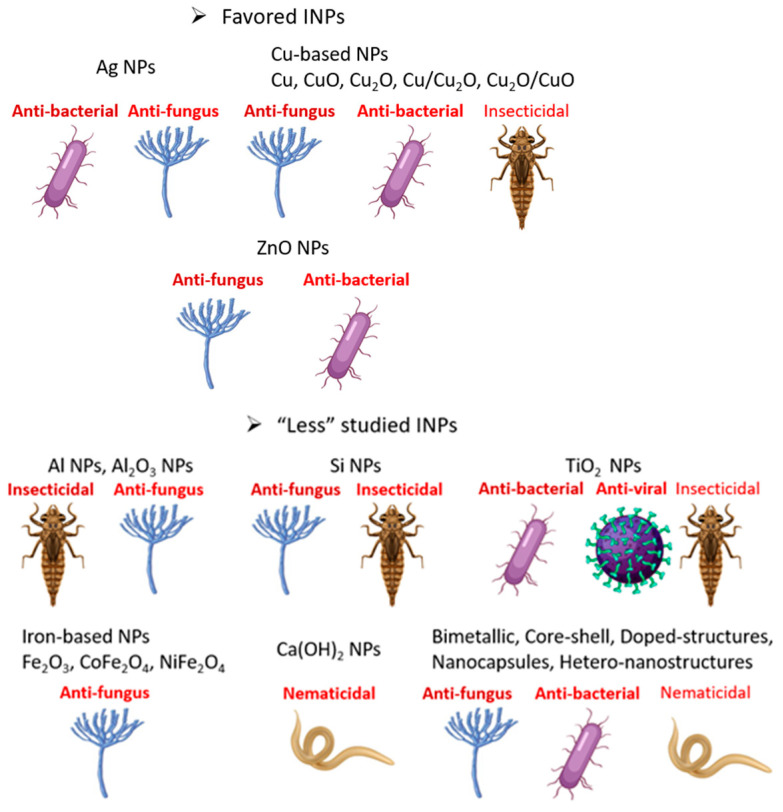
Schematic presentation of the favored INPs and the “less” studied INPs used as anti-phytopathogenic agents and tested in vitro, in pot experiments, and under greenhouse conditions.

**Figure 5 materials-16-02388-f005:**
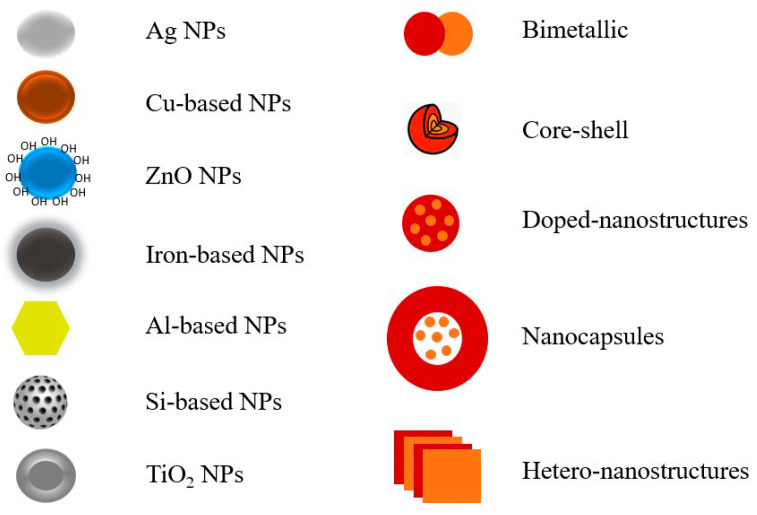
Schematic illustration of advanced inorganic-based nanostructures.

**Table 1 materials-16-02388-t001:** Summary of engineered INPs against classified species (bacteria, fungi, viruses).

	Nanoparticle Properties				Antimicrobial Properties		Ref
Composition	Preparation	Size	Shape	Effective Parameter	Specie	Evaluation Method	
Ag	Chemical reduction	40–60 nm	Spherical	Concentration	*Rhizoctonia solani*	In vitro	[51]
Ag	Chemical/Turkevich	52 nm	Spherical	Concentration	*Phomopsis* spp.	In vitro & *in planta*	[52]
Ag-gelatin	Chemical reduction	5–24 nm	Spherical	Concentration	*Colletotrichum gloesporioides*	In vitro	[53]
Ag-PVP	Modified Tollens’ method	-	-	Concentration	*Sclerotinia sclerotiorum*	In vitro	[54]
Ag	High-voltage arc discharge method	20 nm	-	Exposure time	*Fusarium culmorum*	In vitro	[55]
Ag	Dual reduction	30 nm	-	Concentration, exposure time	*Fusarium* spp.	In vitro	[56]
Ag	Electrolysis	-	-	Application time (before infection)	*Bipolaris sorokiniana*, *Magnaporthe grisea*	In vitro	[57]
Ag	Chemical reduction	7.5 nm	-	Concentration, exposure time	*Gibberella fujikuroi*	In vitro *& in planta*	[58]
Ag-SDS	Direct-current, Atmospheric-Pressure, Glow Discharge (dc-APGD)	28 nm	-	Concentration	*Dickeya* spp., *Pectobacterium* spp., *Erwinia amylovora*, *Clavibacter* *michiganensis*, *Ralstonia solanacearum*, *Xanthomonas campestris*	In vitro	[59,60]
Ag	Chemical reduction	10–100 nm	Spherical	Exposure time	*Ralstonia solanacearum*	*In planta*	[61]
Ag-bovine submaxillary mucin	Chemical synthesis	5–20 nm	-	Size, concentration	*Acidovorax* sp., *Xanthomonas* sp., *Clavibacter* sp.	In vitro, pot experiments	[62]
Ag	Chemical reduction	12 nm	Spherical	Exposure time	*Potato virus Y (PVY)*	*In planta*	[63]
Ag	Commercial	-	-	Concentration, application time (before infection)	*Potato virus Y (PVY)*, *Tomato mosaic* *virus (ToMV)*	*In planta*	[64]
Ag	Co-precipitation	12.6 nm, 8 nm	-	Concentration, application time (after infection)	*Tomato spotted wilt* *virus (TSWV)*, *Bean yellow mosaic virus (BYMV)*	*In planta*	[65,66]
Pegylated Cu_2_O, Cu/Cu_2_O	Hydrothermal	11–55 nm	Spherical	Concentration, composition phase Cu_2_O	*Phytophthora infestans*	In vitro, *in planta* (field exper.)	[70]
Cu, CuO	Commercial	25 nm, <50 nm	-	Concentration, sensitivity in target site	*Botrytis cinerea*, *Alternaria alternata*, *Monilia fructicola*, *Colletotrichum* *gloeosporioides*, *Fusarium solani*, *Fusarium oxysporum*, *Verticillium dahliae*	In vitro *& in planta*	[80]
Cu	Bifunctional molecule- assisted method	50 nm	Spherical	Concentration	*Alternaria solani*	*In planta*	[81]
Cu	Chemical reduction	53 nm	Spherical	Concentration, exposure time	*Aspergillus niger*, *Fusarium oxysporum*, *Phytophthora capsici*	In vitro	[82,83]
Cu	Chemical reduction	345 nm	Polygonal	Shape	*Fusarium oxysporum*	In vitro	[84]
Cu-animal protein, non-ionic polymer, ionic polymer	Modified wet chemistry	5–10 nm	Spherical	Size, concentration, application time (developmental stage)	*Fusicladium oleagineum* *Colletotrichum* spp.	In vitro, *in planta*	[85]
Cu-CTAB	Chemical reduction	3–10 nm	Spherical	Size	*Phoma destructiva*, *Curvularia lunata*, *Alternaria alternata Fusarium* *oxysporum*	In vitro	[86]
Cu-CTAB	Chemical reduction	20–50 nm	Spherical	Concentration	*Fusarium equiseti*, *Fusarium oxysporum*, *Fusarium culmorum*	In vitro	[87,88]
Cu_2_O@OAm, Cu/Cu_2_O@ OAm	Solvothermal	30 nm, 170 nm	Spherical, nanorods	Concentration, composition phase Cu_2_O	*Saccharomyces cerevisiae*	In vitro, *in planta*	[89]
Cu/Cu_2_O@ PEG 8000	Aqueous-phase synthesis	42 nm	Spherical	Concentration, composition phase Cu_2_O	*Fusarium oxysporum*	In vitro	[90]
CuS	Pyrolytic technique	-	Spherical, granular	Shape	*Fusarium* spp.	In vitro	[91]
CuO	Modified wet chemistry	5 nm	Spherical	Concentration, zeta-potential	*Agrobacterium tumefaciens*, *Dickeya* *dadantii*, *Erwinia amylovora*, *Pectobacterium carotovorum*, *Pseudomonas corrugata*, *Pseudomonas savastanoi*, *Xanthomonas campestris*	In vitro	[92]
Cu_2_O@ PEG 8000	Hydrothermal	16 nm	Spherical	Size	*Xanthomonas campestris*, *Escherichia coli*, *Bacillus subtilis*, *Bacillus cereus*, *Staphylococcus aureus*	In vitro	[93]
Cu@ Tween 20	Hydrothermal	46 nm	Spherical	Concentration, metallic core Cu	*Erwinia amylovora*, *Xanthomonas campestris*, *Pseudomonas syringae*	In vitro, *in planta*	[94]
Cu	Chemical reduction	18–33 nm	-	Size, concentration	*Xanthomonas oryzae*	*In planta*	[96]
CuO	Direct precipitation	20 nm	Flower-like	Exposure time, morphology	*Spodoptera littoralis*	*In planta*	[97]
ZnO	Sol-gel	20–35 nm	Spherical	Concentration	*Erythricium salmonicolor*	In vitro	[103]
ZnO	Microwave synthesis	30 nm	Spherical	-	*Fusarium graminearum*	*In planta*	[104]
ZnO	Commercial	-	-	Concentration	*Fusarium graminearum*	In vitro	[105]
ZnO	Colloidal, Hydrothermal synthesis	Diam. 246 nm, Thick. 48 nm	Platelet	Shape	*Fusarium solani*, *Colletotrichum gloesporioids*	In vitro	[106]
ZnO	Controlled precipitation	20–70 nm	-	Concentration, exposure time	*Colletotrichum* sp.	In vitro	[107]
ZnO	Solvothermal	<100 nm	Spheroidal	Composition	*Mycena citricolor*	In vitro	[108]
ZnO	One-pot chemical precipitation	65 nm	Irregular, porous structure	Shape	*Alternaria alternata*, *Fusarium verticilliodes*	In vitro	[109]
Pd or Ce-doped ZnO	Sol-gel, precipitation, microwave-assisted hydrothermal	55–100 nm	Flower-like	Composition, shape, concentration	*Candida albicans*, *Aspergillus niger*, *Aspergillus flavus*	In vitro	[110,111]
ZnO	Commercial	<100 nm	-	Concentration, foliar spray	*Pseudomonas syringae*, *Xanthomonas campestris*, *Pectobacterium carotovorum*, *Pectobacterium betavasculorum*, *Ralstonia solanacearum*	In vitro, *in planta*	[112,113]
ZnO	Chemical/Bath Deposition	-	Nanorods	Concentration	*Pseudomonas syringae*	*In planta*, In vitro	[114]
ZnO	Sol-gel	55 nm	Spherical	Exposure time	*Tobacco mosaic Virus* (TMV)	*In planta*	[115]
ZnO	Commercial	-	-	Concentration, foliar spray	*Tomato Mosaic Virus* (ToMV)	*In planta*	[116]
Fe_2_O_3_	Wet chemistry (green approach)	10–30 nm	Spherical	Concentration, species sensitivity	*Trichothecium roseum*, *Cladosporium herbarum*, *Penicillium chrysogenum*, *Alternaria alternata*, *Aspergillus niger*	In vitro	[118]
CoFe_2_O_4,_ NiFe_2_O_4_	Co-precipitation	25 nm	Spherical	Concentration	*Fusarium oxysporum*	*In planta*	[119]
Al-based	Microemulsion	100–250 nm	Spherical, mesoporous	Concentration	*Fusarium oxysporum*	In vitro, *in planta*	[120]
Si-based	One-pot direct template	20–150 μm	Mesoporous	Concentration	*Alternaria solani*	In vitro, *in planta*	[121]
Al_2_O_3_	Glycine-Nitrate combustion synthesis	10 μm	Amorphous	Concentration, exposure time	*Acromyrmex lobicornis*, *Sitophilus* *oryzae*, *Rhyzopertha dominica*	*In planta*	[122,123]
SiO_2_	Sol-gel	20–60 nm	Spherical	Concentration	*Callosobruchus maculates & Sitophilus oryzae*	*In planta*	[124,125]
TiO_2_	Controlled precipitation	76 nm long, 8 nm wide	Needle	Concentration, foliar spray	*Bactericera cockerelli*	In vitro, *in planta*	[126]
TiO_2_	Commercial	-	-	Concentration, foliar spray	*Xanthomonas* spp.	*In planta*	[127]
TiO_2_-oleic acid	Commercial	3–5 μm	Hollow	Shape, foliar spray	*Broad bean strain virus (BBSV)*	*In planta*	[128]
CuZn-glycol	Microwave-assisted Polyol Process (MW-PP)	20 nm	-	Concentration	*Saccharomyces cerevisiae*	*In planta*	[129]
CuZn-glycol	Solvothermal	35 nm	Nanoflower	Concentration	*Botrytis cinerea*, *Sclerotinia sclerotiorum*	*In planta*	[130]
CuFe- pegyllated	Chemical reduction, Hydrothermal	40 nm	-	Composition (Cu released ions)	*Meloidogyne* spp.	In vitro	[131]
Si-Cu-Quat	Sequential addition, Sol-gel	50–600 nm (silica core), <10 nm (Cu NPs)	Core-shell	Composition	*Xanthomonas perforans*	*In planta*	[133]
ZnO-nCuSi	Sol-gel	600–1100 nm	Core-shell	Composition	*Xanthomonas alfalfae* subsp. *Citrumelonis*, *Pseudomonas syringae pv. syringae*, *Clavibacter michiganensis* subsp. *michiganensis*	In vitro, *in planta*	[134]
TiO_2_/Ag_3_PO_4_, TiO_2_/Cu_2_(OH)_2_CO_3_	Solvothermal, In situ precipitation	2–5 nm of dopants	Microspheres, NPs	Composition, exposure time	*Fusarium graminearum*	In vitro	[135,136]
Ag-doped TiO_2_	Sol-gel	-	Hollow sphere	Visible light exposure duration	*Fusarium solani*	In vitro	[137]
Zn-doped TiO_2_	Commercial	7 nm	-	Concentration, exposure time	*Xanthomonas perforans*	In vitro, *in planta*	[138]
Cu-doped ZnO	Microwave-assisted polyol process	12 nm	Spheroidal	Concentration	*Botrytis cinerea*, *Sclerotinia sclerotiorum*, *Meloidogyne javanica*	In vitro, *in planta*	[139]
ZnO-ZmEO	Precipitation	-	Nanocapsule	Concentration, exposure time	*Alternaria solani*	In vitro	[141]
Cu_2_O/ NiFe_2_O_4_	Solvothermal	30 nm Cu_2_O, 9 nm NiFe_2_O_4_	Spherical	Concentration, exposure time	*Saccharomyces cerevisiae*	In vitro	[142]
ZnO-ZmEO	Precipitation	-	Nanocapsule	Concentration, exposure time	*Fusarium* spp.	In vitro	[140]

## Data Availability

Not applicable.

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
