# Peer review of "Fighting Phytopathogens with Engineered Inorganic-Based Nanoparticles"

_materials, 2023, doi:10.3390/ma16062388_

Round 1

Reviewer 1 Report

this manuscript presents review of research activity concerning the use of nanoparticles as anti-phytopathogens with potential use in agriculture. The authors have done a fantastic job of providing appropriate background to understand the strengths and limitations of traditional pesticides, and the need for new generation pesticides that are both effective, long-lasting, commercially viable, and as safe as practically possible. 

they have provided a good overview of the mechanism of anti-bacterial, anti-viral, anti-fungal, etc. effects of INPs. A good review of published work in each of these areas and a systematic approach to presenting the information makes it much easier to read the review.

This is a well-written review and I do not personally see any important work in this field that is missing. 

One comment for the authors is to include visual content such as figures from individual works cited/mentioned in this review, and graphical representation of the nanopatricle morphologies, etc. Also, figures of mechanistic aspect of their anti-phytopathogenic properties could also be used. As such the text has very few figures, due to which it reads like a huge wall of text. While I understand that nanoparticle and material science research pubs generally contain fewer images, especially compared to organic chemistry works. However, it is also possible to include some valuable images, which will not only add to the depth of scientific descriptions presented in the review, but will also make reading less monotonous.

other than that, the MS is in near publishable quality, and once my suggestion is implemented, it would be fit for publication.

Author Response

Reviewer 1

This manuscript presents review of research activity concerning the use of nanoparticles as anti-phytopathogens with potential use in agriculture. The authors have done a fantastic job of providing appropriate background to understand the strengths and limitations of traditional pesticides, and the need for new generation pesticides that are both effective, long-lasting, commercially viable, and as safe as practically possible. 

they have provided a good overview of the mechanism of anti-bacterial, anti-viral, anti-fungal, etc. effects of INPs. A good review of published work in each of these areas and a systematic approach to presenting the information makes it much easier to read the review.

 This is a well-written review and I do not personally see any important work in this field that is missing. 

One comment for the authors is to include visual content such as figures from individual works cited/mentioned in this review, and graphical representation of the nanopatricle morphologies, etc. Also, figures of mechanistic aspect of their anti-phytopathogenic properties could also be used. As such the text has very few figures, due to which it reads like a huge wall of text. While I understand that nanoparticle and material science research pubs generally contain fewer images, especially compared to organic chemistry works. However, it is also possible to include some valuable images, which will not only add to the depth of scientific descriptions presented in the review, but will also make reading less monotonous.

 Other than that, the MS is in near publishable quality, and once my suggestion is implemented, it would be fit for publication.

Our response: We thank the reviewer for the comments. Based on his/her suggestion we add 3 figures in the revised text; i) Fig. 2: A schematic representation of the NPs’ uptake by plants, ii) Fig. 3: Schematic representation of mechanisms of antimicrobial actions of inorganic-based NPs and iii) Fig. 5: Schematic illustration of the advanced inorganic-based nanostructures.

Reviewer 2 Report

The manuscript has provided an update on the latest advances of engineered inorganic-based nanoparticles for fighting phytopathogens. The results are presented clearly and the conclusions are mostly based on rational arguments. The manuscript can be accepted after major revisions. Before that, please clarify the following comments, which will enhance the quality of the work further.

1. The authors should provide more pictures for vividly illustrating the related research progress.

2. The Table should provide the detailed performances of these engineered inorganic-based nanoparticles for summarizing the characteristics of the various materials.

3. The authors should carefully read the manuscript as it would improve the quality and eliminate the typos.

Author Response

Reviewer 2

 The manuscript has provided an update on the latest advances of engineered inorganic-based nanoparticles for fighting phytopathogens. The results are presented clearly and the conclusions are mostly based on rational arguments. The manuscript can be accepted after major revisions. Before that, please clarify the following comments, which will enhance the quality of the work further.

  1. The authors should provide more pictures for vividly illustrating the related research progress.

Our response: We thank the reviewer for the comments. Based on his/her suggestion we add 3 figures in the revised text; i) Fig. 2: A schematic representation of the NPs’ uptake by plants, ii) Fig. 3: Schematic representation of mechanisms of antimicrobial actions of inorganic-based NPs and iii) Fig. 5: Schematic illustration of the advanced inorganic-based nanostructures.

  1. The Table should provide the detailed performances of these engineered inorganic-based nanoparticles for summarizing the characteristics of the various materials.

Our response: We thank the reviewer for this suggestion. An additional column with the respective preparation method was inserted in the Table.

  1. The authors should carefully read the manuscript as it would improve the quality and eliminate the typos.

Our response: The manuscript has been carefully checked and corrected, to match the journal’s language standards.

Round 2

Reviewer 2 Report

The manuscript can be accepted in present form.